# Identifying Vegetation in Arid Regions Using Object-Based Image Analysis with RGB-Only Aerial Imagery

**Micha Silver \*, Arti Tiwari and Arnon Karnieli** 

Remote Sensing Laboratory, Jacob Blaustein Institutes for Desert Research, Ben Gurion University,
Beer Sheva 84105, Israel; arti@post.bgu.ac.il (A.T.); karnieli@bgu.ac.il (A.K.)
\* Correspondence: silverm@post.bgu.ac.il; Tel.: +972-523-665-918

**Abstract:** Vegetation state is usually assessed by calculating vegetation indices (VIs) derived from remote sensing systems where the near infrared (NIR) band is used to enhance the vegetation signal. However VIs are pixel-based and require both visible and NIR bands. Yet, most archived photographs were obtained with cameras that record only the three visible bands. Attempts to construct VIs with the visible bands alone have shown only limited success, especially in drylands. The current study identifies vegetation patches in the hyperarid Israeli desert using only the visible bands from aerial photographs by adapting an alternative geospatial object-based image analysis (GEOBIA) routine, together with recent improvements in preprocessing. The preprocessing step selects a balanced threshold value for image segmentation using unsupervised parameter optimization. Then the images undergo two processes: segmentation and classification. After tallying modeled vegetation patches that overlap true tree locations, both true positive and false positive rates are obtained from the classification and receiver operating characteristic (ROC) curves are plotted. The results show successful identification of vegetation patches in multiple zones from each study area, with area under the ROC curve values between 0.72 and 0.83.

**Keywords:** segmentation; classification; vegetation; arid regions; gray-level co-occurrence matrix; texture; object-based image analysis; threshold; optimization

## 1. Introduction

As early as 1974, Rouse et al. [1] proposed the well-known normalized difference vegetation index (NDVI), which is based on the difference between the maximum absorption of radiation in the red band (620–680 nm) due to chlorophyll pigments and the maximum reflection of radiation in the near infrared (NIR) band (720–780 nm) caused by leaf cellular structure. With this basic tool, remote sensing has played a key role in vegetation mapping, even in arid regions. For example, a thorough population dynamics study of *Acacia* species (Isaacson et al. [2]) in the arid southern desert of Israel used both ground surveys and NIR band aerial images to follow changes in the canopy cover and tree size distribution. In another early paper, Wiegand et al. [3] compared NDVI from Landsat TM (Thematic Mapper) imagery to a spatial distribution of *Acacia*, also in southern Israel. Both of those research projects analyzed population distributions of *Acacia* by comparing NDVI-derived tree vitality to topography and ephemeral flooding in the dry river beds of their study areas.

More recently, both multispectral and hyperspectral imagery have also been used to identify and characterize vegetation. A review of applications of multispectral and hyperspectral imagery to the mapping of mangrove forests appeared in Pham et al. [4]. They covered spectral-based classifiers as well as object-based image analysis. Both Paz-Kagan et al. [5] and Hong et al. [6] have shown

that hyperspectral images with limited spatial coverage can be used to train multispectral images with a larger spatial extent for vegetation mapping. Hong et al. [6] used small-scale hyperspectral images to train three classification models, then applied the models to multispectral images of a much larger area. Similarly, Paz-Kagan et al. [5] identified the penology stage of an invasive species using hyperspectral data with a random forest classifier, then expanded the analysis to a much larger region using multispectral imagery. Hong et al. [7] addressed the issue of mixed pixels by expanding the classic linear mixed model to more accurately derive abundance maps. They applied their method using both synthetic data and hyperspectral images over an urban region and showed high-quality identification of urban vegetation areas and good separation from non-vegetation pixels.

## 1.1. Vegetation Indices

The NDVI has found extensive use in various applications such as space-time trend analysis of vegetation health (Shoshany and Karnibad [8]), mapping of invasive species (Paz-Kagan et al. [5]), and identification of environmental factors that influence vegetation (Karnieli et al. [9]). Despite widespread adoption, several notable limitations of this index have been documented. For example Mbow et al. [10] critically examined the correlation between NDVI and biomass, measured as above-ground net primary production. Théau et al. [11] compared several different vegetation indices using multispectral satellite imagery and reported inconsistencies between them. Peng et al. [12] reviewed the MODIS NDVI product in the context of "spring greenup" and discovered spatial heterogeneity compared to other vegetation index products. To overcome these drawbacks, alternative vegetation indices (VI) have appeared, and their advantages have been demonstrated. Huete [13] introduced the soil-adjusted vegetation index (SAVI), then a few years later the modified SAVI (MSAVI) was proposed by Qi et al. [14]. Following that work, Huete et al. [15] presented a comparison of the NDVI to another adjusted index: the soil and atmosphere adjusted vegetation index (SARVI). This index, which uses the NIR and red bands after applying atmospheric correction, showed better results in desert regions. Importantly, almost all of these commonly used indices rely on the NIR band to distinguish vegetation.

However, a large archive of aerial imagery is available covering only the visible spectrum, i.e., the three red, green, and blue (RGB) bands. Among the vegetation indices, only a few attempt to differentiate vegetation with RGB bands only. Motohka et al. [16] presented and analyzed the green-red vegetation index (GRVI) on a seasonal time scale. The GRVI is derived similarly to the NDVI:

$$GRVI = (\rho_{green} - \rho_{red})/(\rho_{green} + \rho_{red}), \tag{1}$$

where each $\rho$ component refers to reflectance at a specific spectral band. McKinnon and Huff [17] also tested the accuracy of two RGB-only vegetation indices from drone-acquired images: the visible atmospheric resistance index (VARI) and the triangular greenness index (TGI).

$$VARI = (\rho_{green} - \rho_{red})/(\rho_{green} + \rho_{red} - \rho_{blue}) \tag{2}$$

$$TGI = \rho_{green} - 0.39 * \rho_{red} - 0.61 * \rho_{blue} \tag{3}$$

Results in that work were inconsistent. They reported a good correlation between the RGB-only indices and the NDVI in healthy corn fields, but a less accurate match in rice fields. Their conclusions refer to sporadic matches between these RGB-only indices and the actual crop health. Moreover, desert plants are relatively sparse, their photosynthetic duration is short, and their color is more grey than green, further challenging a pixel-based vegetation index approach.

The recent advancement of drone technology for acquiring aerial imagery has revived interest in RGB-only methods to classify vegetation. The need to use both archived and new consumer-grade, drone-based, RGB-only imagery has led to a different approach. The pixel-based spectral signature classification, which underpins all VIs, is being replaced by object-based image analysis (OBIA).

OBIA took a foothold in the remote sensing discipline and became known as geographic OBIA (GEOBIA), some two decades ago. By 2008, GEOBIA techniques had become a primary tool for image segmentation and classification (Blaschke et al. [18], and Cheng and Han [19]). The advantages of object-based over pixel-based classification were reported and summarized by Myint et al. [20] and Hussain et al. [21]. Feng et al. [22] applied drone images to the mapping of vegetation in an urban environment. In that work, using OBIA techniques, the researchers were able to differentiate trees and grass from the surroundings.

*1.2. Image Texture*

Image texture, a central component of OBIA, describes the relationship between a pixel and it's surrounding neighbors within a given window size. By characterizing this relationship, it becomes possible to distinguish, for example, areas that are homogeneous from areas of high local contrast. Texture parameters are derived from a gray-level co-occurrence matrix (GLCM), first presented by Haralick et al. [23]. Alternative algorithms include wavelet transform, the Gabor transform, Laws energy filter (Laws [24]), and others. A comparison of these different algorithms that appears in Selvarajah and Kodituwakku [25] finds only minor differences in their ability to recognize content in generic images. Ruiz et al. [26] performed a comparison between texture-based and spectral-based classification on satellite imagery using several different texture routines, including GLCM. He analyzed imagery covering three forest and one urban area and found no definitive difference among the texture-based classifications. The GLCM method was also chosen by Marceau et al. [27] in an early work to evaluate SPOT (Satellite Pour l'Observation de la Terre) satellite classification procedures over a mixed urban and forested coastal region of northern Quebec.

The GLCM method examines the relative frequency of pairs of gray-level values for neighboring pixels within a given window size in an image. The matrix is a tabulation of how frequently different combinations of gray levels occur. Aerial photographs usually have 8-bit radiometric resolution, giving a range of 255 shades of gray, thus the co-occurrence matrix is $255 \times 255$ cells. Each cell $(i, j)$ value equals the number of pixels in the image window with value $i$ that have an adjacent cell with value $j$. Furthermore, the GLCM cell values are normalized so that the final matrix contains values from 0 to 1.0.

Once the GLCM is calculated, texture parameters are derived from the matrix. Maillard [28] reviewed eleven GLCM-derived texture parameters and reported that five specific parameters are most often applied in the context of classification of vegetation: angular second momentum, contrast, correlation, inverse distance moment, and entropy.

*1.3. OBIA Applied to Vegetation Classification*

Mapping of vegetation has been specifically targeted by researchers using OBIA. Yu et al. [29] applied OBIA to satellite imagery at 1 m resolution with four spectral bands, RGB and NIR. They used OBIA to create a set of ancillary data, then applied a procedure known as classification and regression tree algorithm (CART) that successfully distinguished different types of vegetation. Lucas et al. [30] applied the proprietary eCognition© software to a time series of Landsat Thematic Mapper (TM) and Enhanced TM (ETM+) imagery to improve the habitat and agricultural area classifications in Wales. Their work utilized both visible as well as infrared bands. OBIA techniques have been applied to the characterization of forests by Blaschke et al. [31]. A work by Cleve et al. [32] also demonstrated a clear improvement in landuse–landcover delineation at urban–wildland interfaces when OBIA was used. Another application of eCognition© software appears in Moffett and Gorelick [33], where they described the advantages of OBIA over the classic pixel-based segmentation methods. They mapped wetland vegetation using 1 m resolution satellite images taking advantage of four bands: RGB and NIR. A study in West Africa (Karlson et al. [34]) applied a multistep procedure to identify tree crowns and clusters. The GEOBIA procedure in their study included classification, OBIA to refine results, and then calculation of NDVI only in those identified pixels to characterize the wooded

areas. Juel et al. [35] performed mapping of vegetation in a coastal region by combining OBIA with a random forest classifier. A report by Alsharrah et al. [36] details a comparison of three vegetation mapping techniques using 2 m resolution satellite imagery in an arid climate: classic VI, OBIA, and a vegetation shadow model. Their results suggest that combining a VI classification with OBIA achieves the best match to true vegetation locations. A large-scale landcover classification was carried out recently by Maxwell et al. [37] by applying GEOBIA to four-band (including NIR) aerial photographs. After creating texture rasters and employing several sets of ancillary data, they reported a very good match between the classification and known objects on the ground.

The majority of research applying GEOBIA to vegetation classification, including all the papers cited above, employed the NIR band. Furthermore, ancillary data layers such as topography (Kim et al. [38]) or LIDAR (light detection and ranging) (Weinstein et al. [39]) were sometimes added as well. In almost all cases, study areas were in a vegetation-rich temperate climate. Significant exceptions are Alsharrah et al. [36] and a study by Ozdemir and Karnieli [40] that focused on forest structure in the semi-arid Negev (see map in Figure 1) desert in southern Israel. Their work, based on multispectral (eight-band) WorldView-2 images, used image segmentation and derived image textures to determine forest structure. That example notwithstanding, almost all previous research with GEOBIA employed spectral bands beyond the visible range and focused on temperate climates.

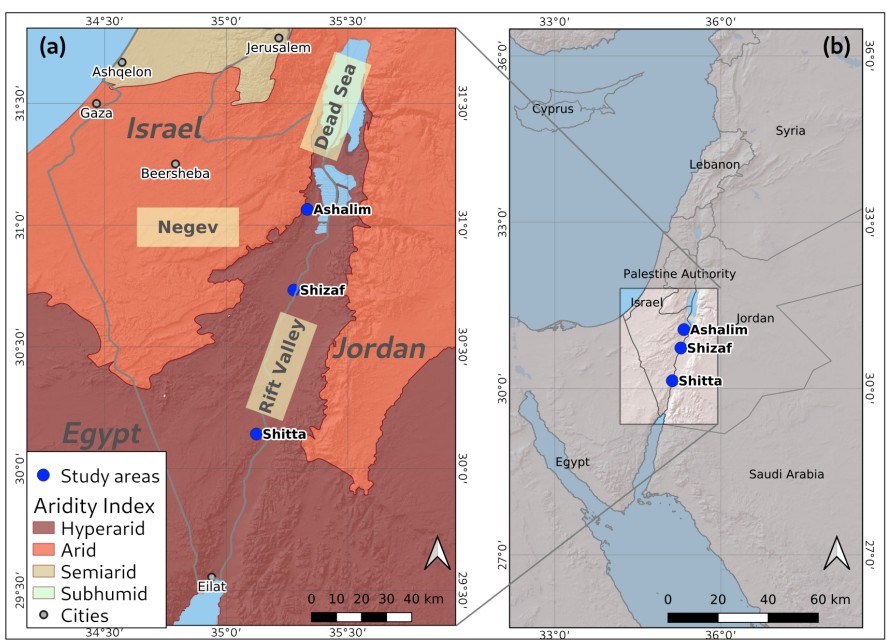

**Figure 1.** (**a**) Three study areas along the Afro-Syrian rift valley with aridity index data from https://cgiarcsi.community/data/global-aridity-and-pet-database/, (**b**) overview map.

The expanded application of GEOBIA was a direct result of improvements in aerial photography. OBIA can discriminate objects and lead to successful classification when the image pixel size is small compared to the object size. With the advent of high-resolution, multispectral aerial photography over the past few decades, application of GEOBIA grew. The importance of pixel size in OBIA is pointed out by Yu et al. [29], and in an older work, Marceau et al. [27] even predicted that with higher resolution images the pixel-based, spectral approach would suffer due to "salt and pepper" effects. In a study applying GEOBIA to drone-acquired imagery for precision agriculture, Torres-Sánchez et al. [41] pointed to spectral heterogeneity as a limitation in classic image classification. OBIA, on the other hand, ideally handles high-resolution imagery. Now that drone aerial images are becoming an accepted research tool with a very small pixel size, GEOBIA techniques are gaining more widespread use.

### 1.4. Segmentation and Classification

OBIA consists of two stages: segmentation and classification. The segmentation stage collects image pixels into clusters such that within each cluster the pixels are alike and between clusters the pixels are different. The measures of likeness and difference, as described by Espindola et al. [42], are: variance within each cluster and spatial autocorrelation between clusters. The balance between these two measures determines how well segmentation identifies real-world objects. If intracluster variance is kept low, then clusters will contain only very similar pixels. This can lead to oversegmentation, where real-world objects become divided and cover several clusters. Conversely, if the spatial autocorrelation between different clusters is maintained low, then intracluster variance increases and one cluster might expand to cover several real-world objects. This balance between intracluster variance and intercluster spatial autocorrelation is regulated by the threshold input parameter (sometimes referred to as scale) to the segmentation procedure. Choosing the best threshold, described by Espindola et al. [42], is crucial for a successful match between segmented clusters and real-world objects.

The classification stage associates each segmented cluster of pixels with a certain class. Many machine learning algorithms, reviewed by Cánovas-García and Alonso-Sarría [43], use supervised classification with a training set of known classes. For example, Yu et al. [29] applied a CART algorithm to identify vegetation in a coastal area of California using image texture rasters derived from the spectral bands, along with ancillary environmental factors. Luca Malatesta et al. [44] compared a maximum likelihood (ML) classifier and a sequential maximum a posteriori (SMAP) model (without OBIA) and reported better results from the SMAP model. Rapinel et al. [45] used an ML classifier together with OBIA to map vegetation in a coastal region of France. They also employed ancillary data and image texture rasters. In a recent work, Mboga et al. [46] applied a fully convolutional neural network to OBIA-derived segmentation to produce landcover maps in an urban setting.

Recent research has often shown a preference for random forest (RF) classifiers (for example, Li et al. [47] and Feng et al. [22]). A comparison of four classification algorithms was presented by Grippa et al. [48], where they performed GEOBIA segmentation and classification in two urban regions. They compared k-nearest neighbors, support vector machine, recursive partitioning, and RF, as well as combinations of the above, and found that RF outperformed all others. A theoretical analysis of RF by Biau [49] and other practical applications (i.e., Nicolas et al. [50], Cánovas-García and Alonso-Sarría [43], Juel et al. [35]) pointed to on-par or superior results compared with the more traditional maximum likelihood or support vector machine classifiers. RF was also applied successfully by Maxwell et al. [37] in a large-scale landcover classification project.

### 1.5. Objectives

This study adopts an object-based image analysis approach for mapping vegetation in arid regions, replacing the traditional pixel-based method that underlies VI calculations. The work attempts to derive an accurate spatial dataset of vegetation patches while restricting the input to the RGB visible bands of aerial imagery to enable full utilization of older, archived photographs as well as consumer-grade, drone-acquired imagery. In addition, recent advancements in GEOBIA are incorporated into the method. This approach to identifying vegetation in a hyperarid region, while limiting the technique to visible bands only, constitutes an innovation.

## 2. Materials and Methods

### 2.1. Study Areas

The GEOBIA technique was applied to three study areas along the hyperarid Rift Valley in southern Israel (Figure 1). These areas were chosen due to the availability of accurate tree locations

from monitoring campaigns. Table 1 lists auxiliary data, in addition to tree locations, that was collected at each study area during the monitoring. The climate in all areas is inducive to a mix of vegetation including some subspecies of *Acacia Vachellia tortilis*, a keystone species in these areas, as well as *Retama raetam*, and *Tamarix aphylla* bushes. These study areas all fall in a hyperarid region with an aridity index, the quotient of precipitation and potential evapotranspiration, below 0.5, as defined by UNEP in the World Atlas of Desertification (Cherlet et al. [51]).

**Table 1.** Auxiliary data collected at each study area during monitoring campaigns.

| Year Initialized | Shizaf 2007 | Shitta 2017 | Ashalim 2012 |
|---|---|---|---|
| Species | x | x | x |
| Number of trunks | x | | |
| Trunk circumference | x | | x |
| Age (est.) | x | | |
| Canopy height (est.) | x | x | x |
| Canopy area (est.) | x | x | |
| Canopy E–W | | | x |
| Canopy N–S | | | x |
| Mistletoe parasite (T/F) | x | | |
| Status (live/dead) | x | x | x |
| Monitoring date | x | | x |
| Continuous Monitoring (T/F) | | | x |
| Flowering | | | x |

The northern site, Wadi (ephemeral riverbed) Ashalim, drains a watershed of approximately 26 km$^2$. The upstream reaches of the watershed are at an elevation of 250 m, and the outlet into the Dead Sea is at $-350$ m. The soil in the upstream region is loess, similar to the desert mountains in southern Israel, whereas near the outlet, the stream bed enters the marl soil that typifies the Dead Sea area. This area is classified in Köppen Geiger as hot semi-arid, with an annual average rainfall of 100 mm and summer daily average temperatures of 41/26 °C (high/low). In addition to the trees and bushes mentioned above, species of *Atriplex* also appear at this site. Analysis was done on a 427 ha area of the hyperarid, lower extent of the wadi, covering three groups of monitored trees.

The Shizaf Nature Reserve was the location of the second study area. With only 40 mm of rainfall per year, this area is classified in Köppen Geiger as a hot desert. The high/low daily summer average temperatures are similar to Ashalim, 41/26 °C. Unlike Ashalim, the nature reserve covers a flat terrain. A large cluster of *Acacia* trees was geolocated in 2005. An area of 396 ha was selected for analysis, which encompassed this group of monitored trees.

The southern site, Wadi Shitta, is located about 100 kilometers further south and drains a small watershed of 16 km$^2$. This wadi exhibits uniform loamy/sandy soil and a moderate slope. The average daily summer temperatures are slightly lower than the northern study areas (40/23 °C) since this wadi is higher in altitude. The ongoing tree monitoring project, part of a Long Term Ecological Research (LTER) site (https://data.lter-europe.net/deims/site/lter_eu_il_015), was carried out in the eastern section of the watershed, but before the wadi enters the marl soil area. Monitoring covers two clusters of some 240 trees (*Vachellia tortilis*, *Vachellia radiana*, and also *Acacia pachyceras*), of which 43 are monitored continuously and 40 others are flagged as dead. The analysis region extended over 256 ha to cover the two clusters of trees.

Aerial Photographs

Ortho-rectified aerial photographs were acquired for each of the regions at a geometric resolution of 25 cm/pixel. The indigenous vegetation in the study areas included trees and bushes with diameters typically above 2 m. Lahav-Ginott et al. [52] report an average tree canopy of 10 m$^2$ in area, and Ward and Rohner [53] refer to 39 m$^2$. The paper by Ward and Rohner [53] included the species *Acacia gerrardii*,

a much larger tree, explaining the difference in canopy size. In either case, tree canopies are covered by at least several tens of pixels in the 25 cm resolution aerial photographs available in the current research. The imagery contained only the visible RGB bands, with eight-bit radiometric resolution, thus each band spanned a gray level range (digital numbers) from 0 to 255. The Ashalim aerial photograph was acquired in 2012. For the Shizaf Natural Reserve study area, an archived aerial photograph was obtained from 2010, some years after the tree monitoring campaign. Since these areas are nature reserves, no substantial changes are expected in the few years between mapping of the tree locations and the acquisition date of the photographs. Photographs from Wadi Shitta were available from 2017, coinciding with the tree monitoring campaign. In all three study areas, the aerial photographs were acquired during the late winter to early spring seasons.

### 2.2. Preprocessing

### 2.2.1. Image Texture

Referring to Figure 2a, five GLCM parameters (Section 1.2) were derived from the green color band of each of the original aerial photographs: angular second momentum, contrast, correlation, inverse distance moment, and entropy. As recognized by Maillard [28], these five, derived with Equations (4) to (8), are most often used in vegetation identification.

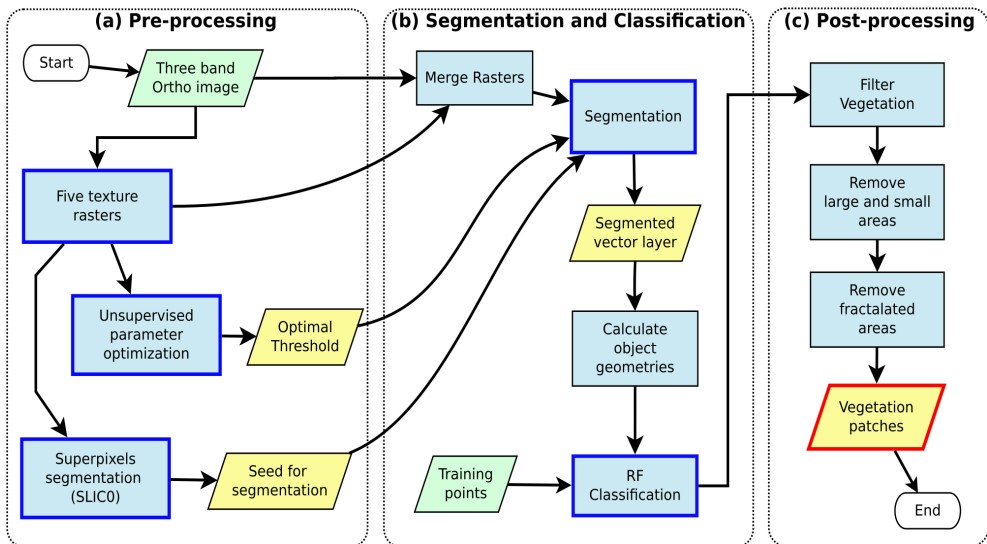

**Figure 2.** Flow diagram of the object-based image analysis procedure.

Contrast:

$$Contr = \sum_{i,j=0}^{N_g} p(i,j)^2;$$

$$(4)$$

Angular Second Momentum (ASM):

$$ASM = \sum_{i,j=0}^{N_g} p_{i,j}^2;$$

$$(5)$$

Entropy:

$$Entr = \sum_{i,j=0}^{N_g} , p_{i,j} * (-ln(p_{i,j}));$$

$$(6)$$

Homogeneity (Inverse Distance Moment):

$$IDM = \sum_{i,j=0}^{N_g} \frac{p_{i,j}}{(1+(i-j)^2)};$$ (7)

Correlation:

$$Corr = \sum_{i,j=0}^{N_g} p_{i,j} * \frac{(i-\mu)*(j-\mu)}{\sigma^2},$$ (8)

where $p_{i,j}$ is the GLCM value at matrix location $(i,j)$, $\mu$ is the mean, and $\sigma$ is the standard deviation of gray-level values within the image window.

GLCM parameter rasters derived from each of the color bands were very similar, and thus were very highly correlated to each other. Including GLCM texture rasters for all colors would have led to overfitting at the classification stage, thus GLCM texture rasters from only one color band (green) were included.

Choice of window size impacts the resulting texture rasters. A small window results in more speckled texture rasters, whereas a larger window smooths the fine texture. A reasonable window size should reflect the smallest object to be differentiated. Considering tree canopies of a few meters (and referring to Lahav-Ginott et al. [52]), in this research a 7 pixel window size (1.75 m) was chosen.

### 2.2.2. Unsupervised Parameter Optimization

Unsupervised parameter optimization (USPO), as introduced by Espindola et al. [42], was implemented by Johnson et al. [54] and Georganos et al. [55]. The routine, applied in this work, performed segmentation repeatedly on small but representative subsets of the original image, while stepping through a range of threshold values. These subset polygons were delineated in advance, ensuring that each subset included a representative mix of the classes in the full analysis area. Then the parameter optimization routine was run in the extent of each subset.

The normalized values of variance and spatial autocorrelation at each iteration were summed (Figure 3), and the optimal threshold was that value that achieved the maximum sum of the two measures.

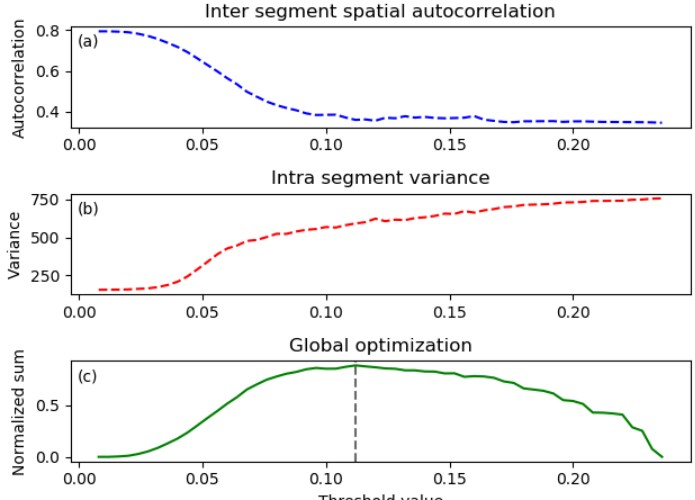

**Figure 3.** Unsupervised parameter optimization. Graph (**a**) shows decrease in spatial autocorrelation between clusters as the threshold increases. Graph (**b**) shows an increase in variance within clusters as the threshold increases. The normalized combination of the two appears in graph (**c**), with the optimal threshold indicated by the vertical dotted line. These graphs were derived from the unsupervised parameter optimization (USPO) procedure in the Shizaf study area.

The optimized threshold for each study area was determined separately since variations in contrast and color balance among the aerial photographs (from different years and different seasons) led to distinct intracluster variance and intercluster spatial autocorrelation for each image. The final optimized threshold values for each study area appear in Table 2.

**Table 2.** Optimal threshold values for each study area.

| Study Area | Optimized Threshold |
|------------|---------------------|
| Ashalim    | 0.11                |
| Shizaf     | 0.13                |
| Shitta     | 0.12                |

### 2.2.3. Superpixels

The concept of superpixels, introduced by Ren and Malik [56], allows for producing a quick preliminary segmentation by k-means clustering. This initial segmentation can be used as a seed for the full segmentation procedure, thus making the overall process more efficient. Among the algorithms for creating superpixels, reviewed recently by Stutz et al. [57], simple iterative linear clustering (SLIC) (Achanta et al. [58]) was shown in that paper to be relatively quick and as successful as the others. An innovative improvement to the SLIC algorithm, known as SLIC0 (pronounced "slick naught"), was demonstrated by Csillik [59]. This approach, implemented in the current work, initializes the regular k-means clustering with a distribution of cluster center points such that nearby center points do not fall on pixels that have similar spectral signatures. In this way, the superpixel clustering ensures that adjacent clusters are different. As shown in Csillik [59], using a seed produced by SLIC0 leads to a final segmentation that stabilizes quickly and more closely matches real-life objects.

### 2.3. Segmentation and Classification

As illustrated in Figure 2b, eight raster layers were used in the segmentation process: the three original color bands and the five texture rasters. These, together with the optimized threshold value and superpixel seed layer as described above, were input into the segmentation routine. The resulting output grouped all similar pixels from the original image into clusters, where each cluster should represent some real-world object. The similarity (i.e., variance) within each cluster and difference (spatial autocorrelation) between clusters was regulated by the threshold parameter. Furthermore, the superpixel preliminary segmentation, used as a seed, allowed the procedure to complete efficiently. By separating the initial raster layers into clusters, this segmentation stage successfully identified real-world objects allowing the following classification stage to correctly associate a class to each cluster. Thus, the segmentation stage was crucial to achieving positive model results overall.

Classification requires, in addition to the segmentation raster output, a dataset of training points. These datasets were prepared manually by on-screen digitization, with the aerial image as background, pinpointing 98, 73, and 74 training points for the Ashalim, Shizaf, and Shitta study areas, respectively. Points were digitized covering trees, sandy areas in the wadi, soil outside the wadi, and rocky areas on the slopes. Care was taken that no tree training point overlapped true tree locations from the monitoring campaigns, thus the validation (Section 2.5) tested tree locations that were kept independent of the training points.

The classification step took into account eleven rasters. First, following the segmentation step, the three initial color bands and five texture bands were used. In addition, three geometric data layers—the area, perimeter, and circle compactness—were prepared for all segmentation clusters. Given a polygon of perimeter $P$ and area $A$, compactness is given by Equation (9):

$$Compact = \frac{P}{2 * \sqrt{\pi * A}}.$$  (9)

All segmentation clusters obtained values for each of these eleven rasters by averaging pixel values within each polygon from each raster. Thus, the classification step modeled one dependent variable, the class, using eleven independent variables. Then classification of the segmented raster was performed using a random forest (RF) classifier. This machine learning algorithm randomly chooses a subset of the independent variables at each tree split, thus making it more resistant to overfitting when variables are correlated. In the current work, variables are mostly derived from the three RGB bands, so correlated variables might be a concern. Therefore, RF was a suitable choice because of both its widespread use (Section 1.4) and avoidance of overfitting. The algorithm was configured with a depth of 200 trees, and number of variables at each split (mtry) of three. Tree depths from 100 to 800 were tested in the Ashalim study area, and a visual examination showed no difference with higher numbers of trees, thus a tree depth of 200 was considered sufficient.

The categorical raster output of the classification procedure assigned to each pixel one of the training category values: vegetation, soil, sand, or rock (Figure 4). In addition, the classification procedure also produced a probability raster, where each pixel was given a value between 0.0 and 1.0 indicating the probability that the pixel should belong to the assigned class. Finally, vegetation patches were obtained by filtering only the the vegetation class from the full classification result, and that filtered raster was vectorized to produce a polygon dataset of vegetation patches.

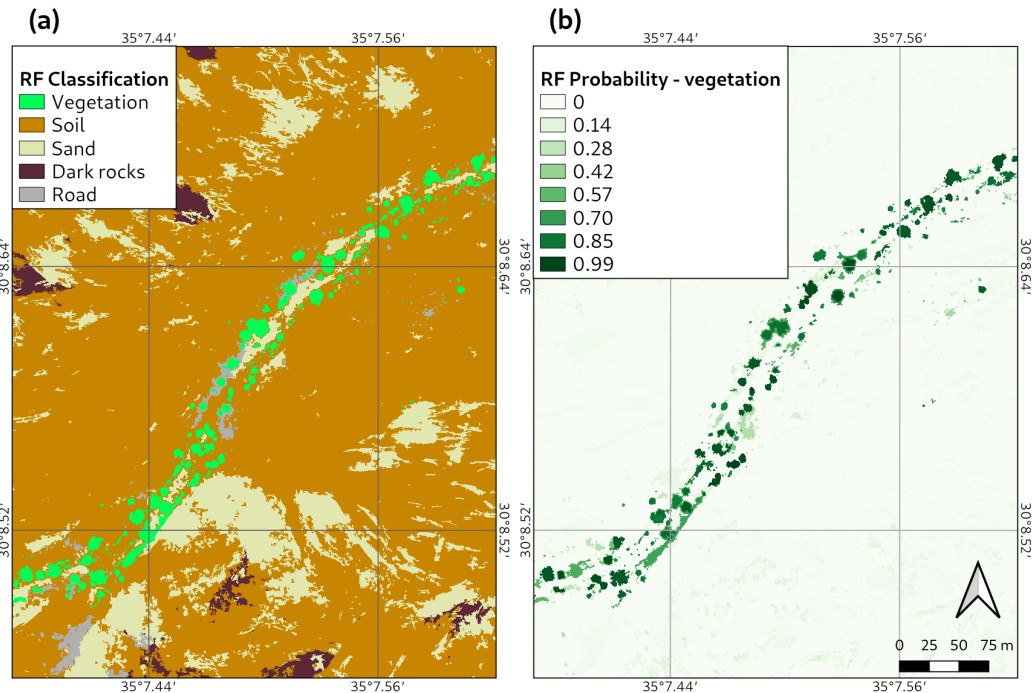

**Figure 4.** Random Forest (RF) classification result (**a**) and RF probability raster (**b**) in a section of the Shitta study area.

## 2.4. Post-Processing

The geometric parameters of area and compactness (Section 2.3) allowed recognition of vegetation patches by their size and shape: long thin areas have a high compactness value. A demographic study of the *Acacia* population, carried out by Lahav-Ginott et al. [52], used panchromatic aerial images to determine canopy cover and tree size distribution. Their work and the study by Ward and Rohner [53] were both based on the assumption of more or less round or oval-shaped tree canopies. They recognized *Acacia* trees in black and white images as darker, circular patches on the light background. However, dead trees and other non-vegetation dark patches do not maintain this round shape. Dead trees appear as very irregular dark shapes, and elongated dark shapes could represent asphalt-paved roads or

shadows under cliffs. Thus, a high circular compactness parameter, as given in Equation (9), was a good indicator of dark shapes that were not vegetation. Using a maximum compactness cutoff value allowed the filtering out of these areas. Furthermore, very small patches were considered suspect and ignored. The empirically determined cutoff values chosen in this work were: maximum compactness = 2.6 and minimum area = 1.0 m$^2$. These steps appear in Figure 2c.

### 2.5. Validation

Validation was carried out in each study area for each group of monitored trees separately. The groups of monitored trees, referred to as validation zones, in all study areas covered only a small portion of the total analyzed area. For example, the analyzed area in the Shizaf reserve extended over 396 ha, while the two validation zones were of 1.8 and 2.5 ha. Analysis was carried out over the full extent in order to visually justify the derived vegetation patches; however, statistical validation was limited to these small zones since true tree locations were available only in the zones. The validation zones surrounding the monitored trees in each zone were delineated by constructing a concave hull (Park and Oh [60]) implemented in R (R Development Core Team [61]) using the concaveman package (Gombin et al. [62]). An example validation zone from the Shizaf study area appears in Figure 5.

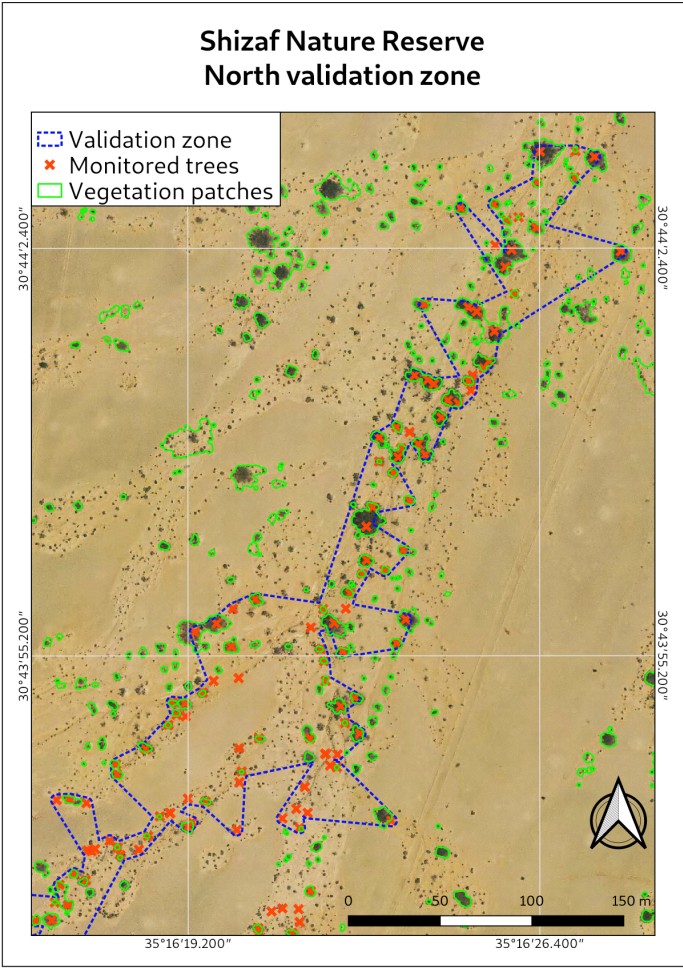

**Figure 5.** Validation zone (blue dashed line) in the Shizaf study area. The monitored tree locations appear as red "Xs".

The number of true tree locations in each validation zone appears in Table 3 (Section 3).

The true trees from the monitoring campaigns were compared to the vegetation patches as determined by the GEOBIA procedure only within each validation zone. Both true positives

(trees correctly identified as vegetation patches) and false positives (vegetation patches that did not overlap true trees) were tallied. In addition, the probability values for all vegetation patches were extracted from the classification probability raster. These probabilities together with the counts of true positives composed the true positive rate (TPR), sometimes called sensitivity. False positives with their probabilities became the false positive rate (FPR), equivalent to $1 - specificity$. The receiver operating characteristic (ROC) curve plots the TPR against the FPR. Then the area under these ROC curves (AUC) represents a measure of accuracy.

*2.6. Implementation*

Application of the method was straightforward. In addition to the three-band ortho-rectified image, preparation of certain vector data was required in advance:

- a small, representative subset of the full study area for USPO;
- a layer of training points for supervised classification (Section 2.3);
- the true tree locations from monitoring campaigns;
- the validation zones as described above in Section 2.5.

The steps described above in Sections 2.2, 2.3, and 2.4 were implemented in the Python scripting language, run within the environment of GRASS-GIS (GRASS Development Team [63]). The choice of this open platform avoids the need for proprietary solutions and allows the details of implementation to be examined and developed further in the future. The code and an example implementation are available from a public repository (https://github.com/micha-silver/obia_vegetation.git). Several GRASS-GIS add-ons (https://grass.osgeo.org/grass76/manuals/addons/) were prerequisite: `i.segment.uspo` and `r.neighborhoodmatrix` for performing the USPO, `i.superpixels.slic` for preparing the superpixel seed and `r.learn.ml`, which contains code for the random forest classifier.

The Python code called GRASS-GIS functions to perform all image analyses and segmentation steps. In the preprocessing stage, these functions created texture rasters, calculated the optimized threshold, and prepared the superpixel segmentation. Then, calls to additional functions performed full segmentation and classification. The classification step was executed with a call to the Python Scikit-learn (Pedregosa et al. [64]) library. This library also included routines for plotting the ROC curves and calculating the AUC as described in Section 2.5.

## 3. Results

The following visual representation of results includes:

- sections of aerial photographs with modeled vegetation and true tree locations;
- graphs showing receiver operating characteristic (ROC) curves;
- a table summarizing AUC values for all validation zones.

The vegetation patches from classification are presented in Figure 6 for one validation zone from each study area. The true tree locations from monitoring campaigns appear in red, and classified vegetation patches are outlined in green. Visual inspection verifies that the GEOBIA procedure successfully located vegetation throughout each study area. Results from Ashalim (Figure 6a) show that some rock faces outside the dry riverbed were incorrectly identified as vegetation. The dark, slightly green shade of volcanic rock covering the hill tops might explain this misidentification. Model results from the Shizaf study area (Figure 6b)show very good identification of vegetation throughout. In the Shitta study area (Figure 6c), some dark patches south of the dry riverbed seem to be missed by the GEOBIA model; however, these are confirmed dead trees and thus correctly skipped, as illustrated in Figure 7. The ground photographs in this figure were taken in late spring, yet clearly, the large tree (panel a) is viable, whereas the tree in panel b shows no vegetation, and the post-processing filter correctly identified this due to the irregular shape of the dead tree.

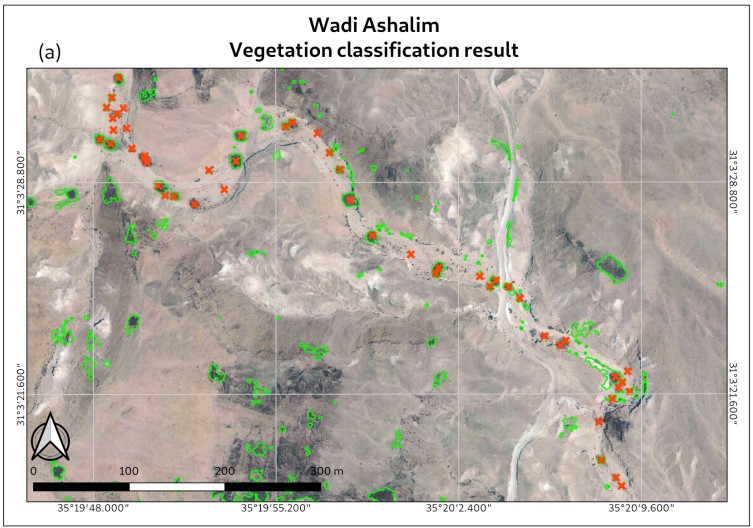

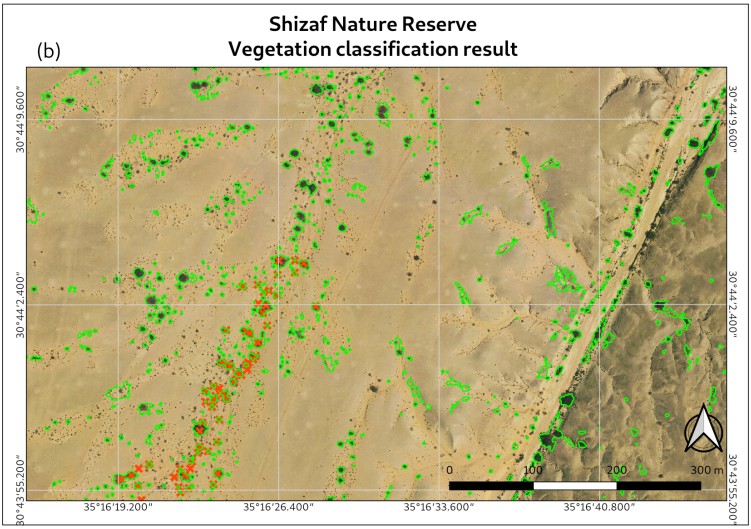

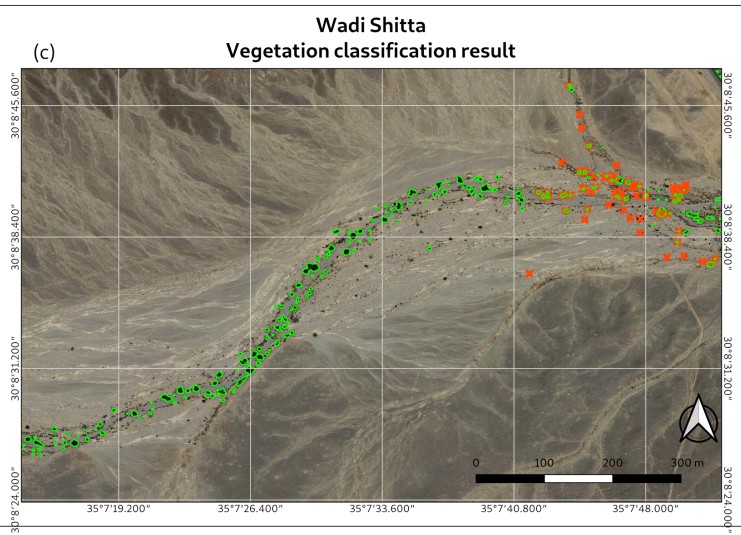

**Figure 6.** Three vegetation classification results. Wadi Ashalim (**a**), northern zone in Shizaf (**b**), and the eastern zone in Shitta (**c**). Monitored tree locations appear as red crosses, and vegetation patches are outlined in green.

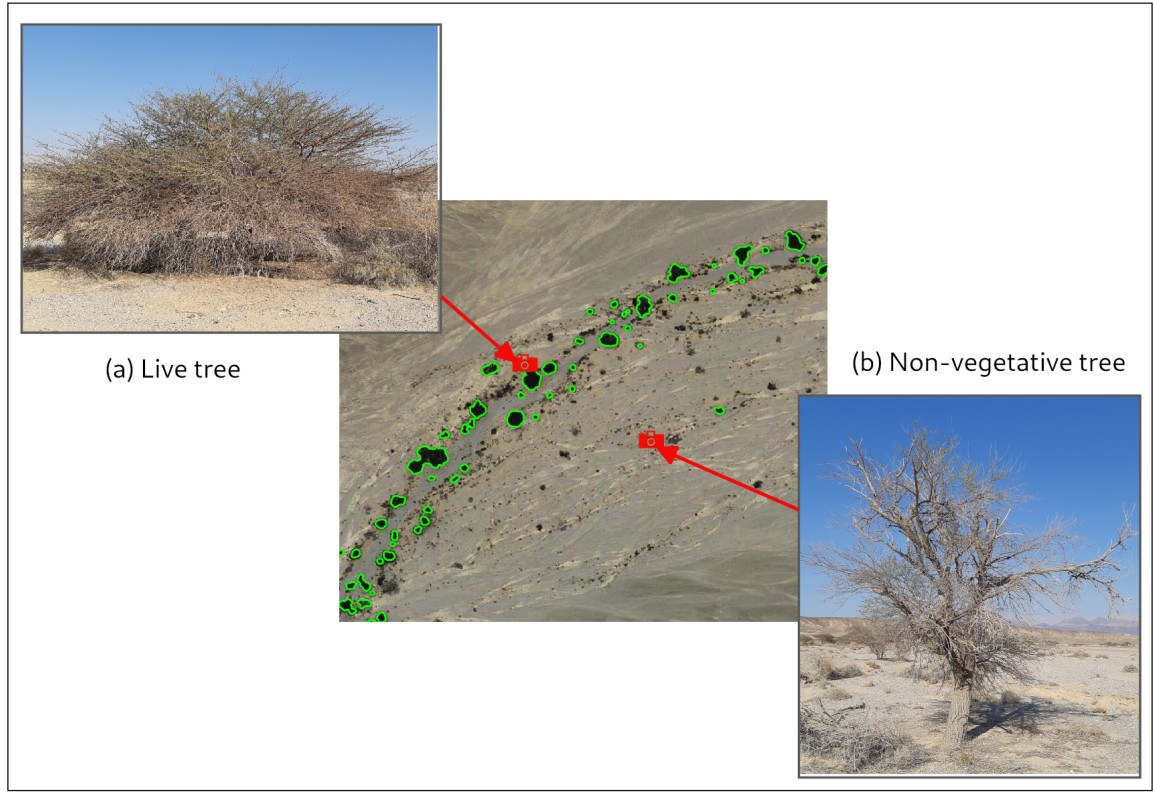

**Figure 7.** Closeup of the Shitta region. Two trees were photographed and the pictures georeferenced. The photo (**a**) shows a live tree correctly identified as a vegetation patch, while photo (**b**) shows a non-vegetative tree that was correctly skipped by the model.

Three sample ROC curves appear in Figure 8, and the complete set of AUC values for all validation zones is presented in Table 3. The northern validation zone in the Shizaf nature reserve, showing the lowest AUC value, encloses many small bushes, especially *Tamarix aphylla*. However, the tree monitoring campaigns all focused on *Acacia* trees, the keystone species in this desert region. Thus, the model correctly identified vegetation patches that were not located in the monitoring campaign, leading to a somewhat high false positive rate and thus a lower AUC.

**Table 3.** Area under the curve (AUC) values and number of validation trees for all validation zones.

| Study Area | Validation Zone | AUC | Number of Trees |
|---|---|---|---|
| Ashalim | Wadi Amiaz | 0.818 | 62 |
| Ashalim | Wadi Ashalim | 0.749 | 85 |
| Ashalim | south | 0.850 | 66 |
| Shizaf | north | 0.712 | 134 |
| Shizaf | south | 0.731 | 159 |
| Shitta | east | 0.830 | 72 |
| Shitta | west | 0.730 | 82 |

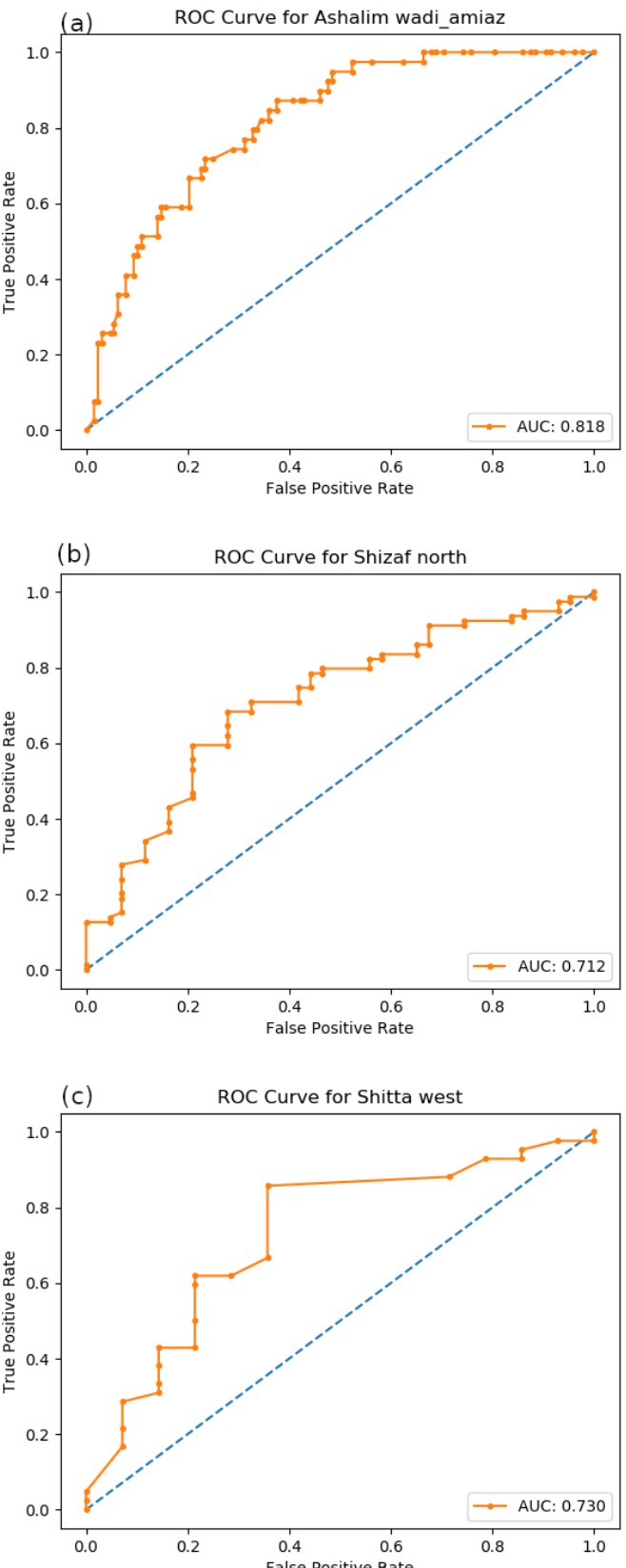

**Figure 8.** Three receiver operating characteristic (ROC) curves. Wadi Amiaz zone in Ashalim (**a**), northern zone in Shizaf (**b**), and the western zone in Shitta (**c**).

## 4. Discussion

Since the beginning of high-resolution, commercial, color aerial photography decades ago, a large archive of imagery consisting of only the RGB visible bands has accumulated. Recently, with the expansion of consumer-grade drone aerial photographs, even more imagery covering only the RGB bands has become available to environmental research. Numerous papers, reviewed briefly in Rapinel et al. [45], have used remote sensing for vegetation mapping. In research similar to the current work, Staben et al. [65] used high-resolution aerial imagery in an OBIA procedure to determine woody biomass in arid and semiarid regions of Australia. Shoshany and Karnibad [8] also examined biomass change and water-use efficiency in the semi-arid region of the eastern Mediterranean. However, each of these efforts used remote sensing data that included an infrared band. Much research has focused on time series analyses of desertification or forest decline (Peters et al. [66], Joshi et al. [67], and more recently Dorman et al. [68], Bajocco et al. [69], Fensholt et al. [70], and Zhang et al. [71]). Remote sensing data has aided research in tracking the health and distribution of certain species of vegetation (Escobar-Flores et al. [72], Pham et al. [4], Paz-Kagan et al. [5]). Yet, again, all of the above employed additional spectral bands.

However, as was demonstrated above, the classic pixel-based classification that takes into account only the spectral signature of the color bands achieves unsatisfactory results. The VI methods were shown to be especially unsuited to arid regions (i.e., Moleele et al. [73]) due to the weak reflectance of green and strong interference from the surrounding bright soil. Mbow et al. [10], working in the semi-arid Sahel region, showed only limited success in vegetation mapping, and only when they used soil moisture as an auxiliary variable.

GEOBIA has become a standard tool in remote sensing for over a decade. By first segmenting an image based on OBIA factors, including image texture, spectral signature, and geometry, real-world objects are correctly separated. Then the second classification stage successfully identifies and classifies those objects. The demand to take advantage of RGB-only aerial imagery has reinforced the move to GEOBIA. Not only does GEOBIA overcome the shortcomings of VI methods, but it also deals very well with the high-resolution imagery available recently by avoiding the "salt and pepper" problem.

The procedure in this work demonstrated successful mapping of vegetation in arid regions, using imagery with only RGB color bands. Initially, five texture rasters were prepared using the GLCM algorithm from one of the color bands. Two innovative preprocessing steps were adopted: a superpixel preliminary segmentation and optimized selection of the threshold parameter. With those inputs, segmentation was executed followed by the classification step using a random forests classifier. The map images and tables presented in Section 3 suggest that accurate mapping of vegetation in arid regions by RGB-only imagery is achievable. The weak green coloring of desert vegetation is overcome by using OBIA texture factors and careful selection of the threshold parameter in segmentation. Furthermore, by adding the geometric measures of area and circle compactness before classification, the model filtered out clusters with irregular or elongated shapes that could not be vegetation patches.

## 5. Conclusions

The GEOBIA remote sensing tool demonstrated in this research can open the way to ecological investigation that was not easily achievable previously by utilizing archives of aerial imagery. Large-scale mapping of vegetation in arid regions potentially raises questions of tree canopy density, change detection, patch analysis, comparisons with explanatory environmental variables, and so on. Ground-based monitoring campaigns can cover only limited areas, so these avenues of research were mostly closed. Early applications of remote sensing, when based on classic vegetation indices, showed limited success in extensive mapping of trees in desert regions. Whereas by adopting and tuning the object-based method presented here, ecologists can obtain relatively accurate vegetation maps both from past archives of RGB-only aerial imagery and from new and inexpensive images acquired by drones. The current work, which applies recent advances in GEOBIA (Subsection 2.2),

could revive ecological research in arid region vegetation by enabling use of archives of RGB-only aerial photographs, merged with recently acquired imagery from consumer grade drones.

The procedure (Section 2) does not require costly proprietary software, rather the steps are transparent and open to critical analysis. The authors believe that with careful testing and adjusting of the threshold parameters, highly reliable vegetation maps can be attained. Looking forward, application of the techniques offered herein could expand research and the understanding of arid region ecosystems.

**Author Contributions:** Conceptualization, M.S., A.K., and A.T.; methodology, M.S.; software, M.S.; validation, M.S.; writing—original draft preparation, M.S.; writing—review and editing, A.K., A.T.; supervision, A.K.; funding acquisition, A.K.

**Funding:** The project leading to this research was partially funded by the Jewish National Fund (JNF) contract no. 10-02-002-17 and by the European Union's Horizon 2020 Research and Innovation Program under grant agreements no. 641762 (Ecopotential) and no. 654359 (eLTER).

**Acknowledgments:** Several actors helped in preparation of this research by supplying the locations of trees from the ground surveying campaigns. Without this data, validation of the technique would not have been possible. The head of the Arava Dead Sea Science Center, Dr. Eli Groner connected the authors with their field staff to obtain tree locations in Wadi Shitta. In addition, Asaf Tsoar and Rotem Golan from the National Parks Authority oversee monitoring of the vegetation in Wadi Ashalim, and they willingly contributed their dataset with the help of their GIS department and Tal Polak. Many thanks to all parties involved for their assistance.

**Conflicts of Interest:** The authors declare no conflict of interest.

## Abbreviations

The following abbreviations are used in this manuscript:

| | |
|---|---|
| MDPI | Multidisciplinary Digital Publishing Institute |
| OBIA | object-based image analysis |
| GEOBIA | geographic object-based image analysis |
| NDVI | Normalized differential vegetation index |
| VI | Vegetation index |
| NIR | near infrared |
| LIDAR | light detection and ranging |
| GLCM | gray-level co-occurrence matrix |
| RF | Random forest |
| RGB | red, green, blue |
| SLIC | simple iterative linear clustering |
| TPR | true positive rate |
| FPR | false positive rate |
| ROC | receiver operating characteristic |
| AUC | area under the curve |

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
