# Peer review of "Identifying Vegetation in Arid Regions Using Object-Based Image Analysis with RGB-Only Aerial Imagery"

_remotesensing, doi:10.3390/rs11192308_

Round 1
Reviewer 1 Report
Please see the attachment.

Reviewer 2 Report
The manuscript presents a step by step a methodology to derive vegetation patches in arid regions. Authors use the color bands and texture metrics of RGB-only aerial imagery in a GEOBIA process, adjusting the threshold parameters. Classification process performed with Random Forests classifier and three geometric parameters.
The introduction is long, and actually it constitutes a review of similar methodologies and work done. However, the aim, as well the importance and novelty of the research conducted should be clearly stated in introduction part. The discussion is short, some comments could be done comparing the results with previous studies
Line 29 The point of these chapter is not totally clear. The chapter is not fully demonstrating the limitation of VI for mapping vegetation in arid regions. You may consider to rename the subtitle to Vegetation indices
Line 30 Various applications you could name some
Line 30 Limitations you could name some
Line 64 Lowercase for the reference HARALICK
Line 67-70 The authors provide a reference of work done. I would recommend adding some information about the environment where those researches were done
Line 79 You should better present all the equation in methodology section than in the introduction.
Line 141 Could you make any reference to similar works for arid regions? And please note what data they used.
Line 197-198 The phrase needs to be explained. What is the average tree area? Can you assume why there is such a big difference between the two authors?
Line 195-205 You should separate the paragraph to a sub-chapter as Aerial imagery or rename the subtitle to Study area and aerial imagery
Line 195-205 It is not referred where the photographs were acquired from. In which scale the aerial photographs were taken and what where the characteristics of the camera.
Line 234 Correct an to as
Figure 3 Refers to which area?
Line 260 What is the number of variables available for splitting at each tree node (mtry)?
Line 264 Could you please present the categorical raster output or a probability raster?
Line 272 Specify the geometric parameters
Line 293 Please refer the package you used
Line 316 please correct add-ons
Reviewer 3 Report
The study focuses on identifying the vegetation in arid regions using object-based image analysis with RGB aerial imagery and using VIs and GLCM features.
Comments:
166- What is the source of the identification of different species loated in the study area including Acacia, Retama, Tamarix, Atriplex, and especially the different species of Acacia which is the major dominant species in the study area?
201- What is the number of images (RGB-photographs) used in each study location? and in which season?
206- Provide a table with specifications of the remote sensing data (The RGB-photographs) used in the current study including the number and dates of the images?
217- Why did you use 7x7 window size? you mentioned that considering tree canopies of a few meters and referred to Lahv-Ginott et al., 2001. do you think the size of the tree canopies did not change from 2001 to 2019?
257- grids or raster layers (245)? It is better to use one term.
261 - Why did you use RF classifier? you have to provide more information.
295- Provide more information about the monitoring campaigns data used for validation? what they measured? how and when?
308- what is the percentage of training to validation data used in the current study?
325- What is the accuracy of the classification in the three different study locations?
325- The Results section is very short.
325- Figure (6) was not mentioned in the text.
- In figure (6) both trees are dry (the live tree and the non-vegetative tree) but in the RGB-photograph the live tree is dark in color, this is the reason to show in which season the RGB images were captured.
341- The Discussion section is very short, it has to be extended and compared with other studies.
Round 2
Reviewer 2 Report
The authors successfully addressed all the concerns and comments
Reviewer 3 Report
Accept in present form